# Polysaccharides for Biodegradable Packaging Materials: Past, Present, and Future (Brief Review)

**DOI:** 10.3390/polym15020451

**Published:** 2023-01-15

**Authors:** Kristine V. Aleksanyan

**Affiliations:** 1Semenov Federal Research Center for Chemical Physics, Russian Academy of Sciences, ul. Kosygina 4, Moscow 119991, Russia; aleksanyan@chph.ras.ru or aleksanyan.kristine@gmail.com; 2Engineering Center, Plekhanov Russian University of Economics, Stremyannyi per. 36, Moscow 117997, Russia

**Keywords:** polysaccharides, biodegradability, cellulose, starch, chitin and chitosan, biodegradable packaging materials

## Abstract

The ecological problems emerging due to accumulation of non-biodegradable plastics are becoming more and more urgent. This problem can be solved by the development of biodegradable materials which will replace the non-biodegradable ones. Among numerous approaches in this field, there is one proposing the use of polysaccharide-based materials. These polymers are biodegradable, non-toxic, and obtained from renewable resources. This review opens discussion about the application of polysaccharides for the creation of biodegradable packaging materials. There are numerous investigations developing new formulations using cross-linking of polymers, mixing with inorganic (metals, metal oxides, clays) and organic (dyes, essential oils, extracts) compounds. The main emphasis in the present work is made on development of the polymer blends consisting of cellulose, starch, chitin, chitosan, pectin, alginate, carrageenan with some synthetic polymers, polymers of natural origin, and essential oils.

## 1. Introduction

Interest in biodegradable packaging materials is dictated by the necessity to decrease the impact of oil-based nondegradable plastics on the environment. The production of the latter, as well as the amount of plastic waste that cannot be collected, utilized, or recycled, is rising every year [1,2,3,4,5,6,7]. Obviously, the biodegradable “competitors” are more expensive and their production requires advanced technological processes, but the benefits for nature and the future sustainable world are more important. The components imparting biodegradability should be first from renewable natural resources that are accessible and low-cost [8,9,10,11,12,13]. One of the classes of such compounds is polysaccharides (Table 1). They meet all the requirements listed above. However, the consumer characteristics can deteriorate when polysaccharides are used. That is why the researchers all over the world propose more and more formulations that can be used as biodegradable packaging materials, for example [14]. The first mention of the biodegradable composition (a starch-based one) was in the early 1970s [15,16]. This approach was directed to utilization of a large-tonnage synthetic polymer polyethylene via introducing a biodegradable component. Apart from this approach, this review will consider other ones involving main polysaccharide-based systems.

It is necessary to note that among all directions in production of biodegradable polymer packaging materials, there are a few that are most widespread: production of compositions based on large-tonnage synthetic polymers and natural polymers; production of compositions based on only natural or naturally occurring polymers; and introduction of molecules carrying functional groups that promote the accelerated photodegradation into biodegradable polymers [17].

Ghanbarzadeh et al. [18] illustrated that nano-scale polysaccharides have great potential to be used in food packaging since they can impart a large range of improved properties (e.g., increase water vapor barrier properties).

It should be noted that there is a whole cluster of investigations dedicated to encapsulation of essential oils (hydrophobic component) into polysaccharides (starch, dextran, chitosan, pectin, etc.) to produce biodegradable food packaging materials [19]. Apart from the described polysaccharides, soluble soybean polysaccharide has also attracted considerable attention from researchers to develop biodegradable packaging materials [20,21]. It was demonstrated that the introduction of essential oils into soybean polysaccharide and production via emulsification allows one to impart antioxidant properties and improve water vapor permeability. The presence of a single *T_g_* indicates the compatibility of the components.

The production of “smart biodegradable packaging” systems with naturally derived pigments from food and food wastes (curcumin, chlorophyll, quercetin, etc.) with polysaccharides as matrices can offer future prospects for developing biodegradable packaging materials with visual quality evaluation [22,23]. Owing to such pigments (changing of their color), one can follow different indicators such as pH, CO_2_ sensitivity, antioxidant, etc. Therefore, Khanjanzadeh and Park showed [24] that cellulose acetate as a matrix with bromocresol purple immobilized on cellulose nanocrystals can serve as a pH-sensitive packaging material. At the same time, the application of neutral red immobilized onto cellulose nanofibrils in poly(vinyl alcohol) matrix allowed one to create pH- and ammonia-sensitive films [25], and to develop ammonia-sensitive materials based on TEMPO-oxidized cellulose nanofibril-based hydrogels with anthocyanins [26].

Since this theme is comprehensively studied and corresponding multiple reviews have been reported, the present work will not cover the polysaccharide sources, syntheses, structures, and application fields (only a common scheme is presented in the work to summarize some information (Figure 1)). The focus of this review is on biodegradable packaging materials of polysaccharides with some popular synthetic polymers, natural and naturally occurring polymers, some essential oils, and plant extracts as a new trend in materials science.

## 2. Polysaccharides and Their Derivatives as Components of Biodegradable Packaging Materials

### 2.1. Cellulose and Its Derivatives

As is known, cellulose is the most widespread polymer in nature. It is a main component of cell walls of higher plants; its content in them depends on the plant type. Cellulose has been used in industry since the 19th century. Interest in cellulose and its numerous derivates is caused by the set of unique properties that can be used in different application fields, including the packaging industry. Obviously, the first approach given above is the most studied for cellulose-based materials. Therefore, a lot of research groups have been interested in the creation of biodegradable systems of cellulose with large-tonnage synthetic polyolefins, namely, polyethylene for example [27,28,29,30,31,32,33,34,35,36,37,38,39,40]. In [27,28,29,30,31], a collective of authors developed biodegradable compositions based on cellulose/ethylcellulose and low-density polyethylene. Investigation of mechanical, thermal properties, and biodegradability showed that varying the content of polysaccharide one can obtain the materials with optimal properties. Therefore, in order to improve these characteristics, another polysaccharide (chitin/chitosan) or poly(ethylene oxide) was introduced [32,33,34,35]. It was demonstrated that with preservation of good mechanical properties, the biodegradability was improved (intensified) compared to binary cellulose-polyethylene blends. Shumigin et al. [36] produced polyethylene–cellulose blends in a twin-screw extruder; their mechanical and rheological properties were compared to those for cellulose-poly(lactic acid). It was revealed that viscosity and moduli of filled polymer increased with the growing amount of cellulose.

Poly(vinyl alcohol) is also used for the creation of cellulose/cellulose derivative-based compositions [41,42,43,44,45,46,47]. Therefore, in [41], poly(vinyl alcohol)-starch compositions were reinforced by cellulose nanofiber. The systems were obtained by the solution casting method. 4 wt% loading of cellulose nanofibers into the compositions led to improvement in thermal properties, as well as tensile strength and elongation at break. The introduction of lemongrass essential oil resulted in antibacterial capacity towards *S. aureus*. Dey et al. [45] showed that the materials consisting of poly(vinyl alcohol), cellulose nanocrystals, and chitosan nanoparticles display high mechanical strength and are biodegradable in nature; thus, they can be considered as potential biodegradable packaging materials.

Naturally occurring polymers (polyesters) are of special interest in the development of biodegradable materials owing to their biodegradability (under certain conditions) and properties similar to those of synthetic polymers. This explains the special attention given to the production of new materials based on them with polysaccharides, cellulose/cellulose derivatives [48,49,50,51,52,53,54,55,56,57,58,59]. Khan et al. [59] produced biodegradable compositions based on polycaprolactone and methylcellulose—a water-soluble derivative of cellulose. Their mechanical, barrier, and degradation properties were defined. The test on degradation was carried out for 6 weeks in aqueous medium. It was established that composites slowly lose their masses in time.

The biodegradable compositions of cellulose and derivatives with other polysaccharides are described in numerous works, for example, [34,35,60,61,62,63,64,65,66,67,68,69,70,71]. All the mentioned works describe in detail the properties and possible application areas for cellulose-based compositions. Therefore, Riyajan [61] used modified cellulose (by lactic acid) with cassava starch. The resulting material possessed improved water resistance and mechanical properties; the composition transparency decreased as the modified cellulose content increased. The tests with banana preservation showed that the film proposed kept it without black/yellow spotting for up to 7 days. Cellulose was also often combined with chitin/chitosan. The correlation concentration/mechanical properties were carried out, and the optimal composition was found [68]. It was established that the introduction of nanocellulose into chitosan led to the improvement in mechanical and water vapor barrier properties. Ghalehno and Yousefi [66] mixed four types of cellulose nanomaterials obtained from wheat straw with carboxymethyl cellulose. The mechanical properties and transparency were taken as the comparative parameters: composites with TEMPO oxidized cellulose nanofibers showed the better results. Namely, their transparency comprised 56%, while for neat carboxymethyl cellulose it was 65%. Such a high value was explained by the smaller size and higher homogeneity of TEMPO-oxidized nanofibers. Higher mechanical characteristics were explained by less dimension and higher specific surface area for TEMPO-oxidized compared to other types of cellulose nanomaterials [66]. Rajeswari et al. [72] suggested the production method for cellulose acetate biodegradable materials with the introduction of other polysaccharides (alginate and carrageenan) in recent work. In [64], crystalline nanocellulose was mixed for the first time with polysaccharide pectin. Improvements in tensile strength and water vapor permeability, as well as reduction in moisture absorption, were achieved.

Proteins, as a large class of natural polymers, were also used to produce biodegradable materials with cellulose [73,74,75,76,77,78]. da Silva Filipini et al. [77] proposed using methylcellulose with collagen or whey protein to produce biodegradable materials by the casting method. Thanks to unique properties of methylcellulose, the resulting materials demonstrated improved mechanical, barrier, and thermal properties with the preservation of transparency at a high level, along with fast biodegradation (10 days). Authors proposed using these materials for the production of soluble sachets for powdered foods, as well as oil containers and capsules for instant coffee machines.

One more “attractive” component gaining particular interest is essential oil used as an antimicrobial agent [79,80,81,82,83,84]. dos Santos Acosta et al. [83] used cinnamon and listea cubeba essential oils to modify the properties of methylcellulose films; an increase in antibacterial activity against S. aureus and E. coli was observed, and the materials biodegraded in black sand and beach soil for 20 days. These oils can promote the production of active packaging with antibacterial capacity.

It should be noted that this theme has been intensively explored in the last decade, thus it is an emerging and promising technique for obtaining novel classes of biodegradable materials.

### 2.2. Starch

Starch is one of the most popular polysaccharides used in the packaging industry, mainly due to its low price. However, starch, as well as other polysaccharides, displays biodegradability, film-forming ability, renewability, non-toxicity, etc.

As was mentioned above, the first biodegradable compositions proposed for application as biodegradable packaging materials were based on starch mixed with polyethylene [15,16]. Following that, this exploration trend was developed in many investigations for example [27,28,33,85,86,87,88,89]. In common sense, the introduction of starch into polyethylene leads to deterioration in tensile strength, elongation at break, and a rise in elastic modulus due to incompatibility of starch and polyethylene [86]. However, the application of plasticized starch (mainly water or glycerol) can eliminate such drawbacks. The biodegradability of starch-polyethylene blends depends on the starch content [27]. The optimal content of starch was proposed to obtain materials with good mechanical characteristics along with biodegradability.

One more polymer often mixed with starch is poly(vinyl alcohol) [90,91,92,93,94]. Priya et al. [92] studied the main characteristics of the starch/PVA compositions in detail, with citric acid as a plasticizer and glutaraldehyde as a cross-linking agent. Increased tensile strength and swelling degree were established. Thermal and antibacterial studies revealed that the synthesized blend films could be used as potential materials in food packaging.

Numerous works are dedicated to investigation of the materials based on starch/thermoplastic starch with polyesters (PLA, PCL), since both components are natural or naturally occurring for example [95,96,97,98,99,100,101,102,103,104,105,106,107,108,109,110,111,112]. Dubois and Narayan [106] proposed using reactive melt processing to obtain biodegradable compositions based on PLA and PCL with corn starch after adequate compatibilization. Controlling the starch content, one can obtain the materials with improved ultimate mechanical properties and accelerated biodegradability. Kulkarni and Narayan [107] declared that the application of maleated thermoplastic starch as a nucleating agent for PLA led to an increase in the rate of crystallization, crystallinity degree, and enhancement in oxygen barrier properties and water vapor permeability with preservation of the PLA biodegradability. Thus, the maleated thermoplastic starch/PLA compositions possess improved barrier and crystallization properties, but similar mechanical, thermal, and biodegradation properties compared to neat PLA. In a series of works, Rogovina et al. [95,96,97,98,99] developed compositions of PLA with potato starch. The compositions were obtained in a Brabender mixer under the action of shear deformations. Introduction of starch to PLA led to some deterioration of the mechanical properties, while plasticization of PLA with poly(ethylene glycol) resulted in a rise in elongation at break. The biodegradability was investigated by independent methods. The results obtained showed that the biodegradation intensity increased with the rise in starch content. The most pronounced biodegradability proceeded in the plasticized systems. Ninago et al. [109] revealed good compatibility between starch and polycaprolactone due to low concentration of the latter. This also resulted in good processability of plasticized starch-polycaprolactone blends, and the properties of the final material appeared to be improved (namely, water vapor barrier capacity and water solubility) without changing the thermal stability. An increase in opacity and UV absorption capacity promoted by polyester blocking effect was detected in the work. Thus, the creation of starch-based compositions with polyesters is a promising approach for the development of biodegradable packaging materials.

One of the largest research fields is the development of biodegradable materials based on starch with other polysaccharide(s). In order to obtain antimicrobial packaging materials, starch was mixed with chitin/chitosan [34,35,113,114,115,116,117,118,119,120,121,122,123]. The solution casting method was used to produce edible corn starch-chitosan film possessing antimicrobial properties [118], which can prevent food spoilage. The mechanical and thermal properties of thermoplastic starch with chitin were studied in [113]. The introduction of chitin as a reinforcing agent resulted in a great rise in elastic modulus and decrease in elongation at break. Thanks to hydrophobicity of chitin, the resulting materials had improved water resistance up to 20% compared to neat starch. The same changes in mechanical characteristics were declared in [117,119], which is connected to the fact that chitin/chitosan are more rigid polysaccharides than starch. In a study of biodegradability, antimicrobial properties are an important part of the investigations in this field [124]. Thus, Lopez et al. [117] stated that the introduction of chitosan into thermoplastic starch reduced growth of *S. aureus* and *E. coli* at contact. One more polysaccharide used in blends with starch for production of biodegradable materials is pectin [125,126,127,128]. These polymers obtained under reactive extrusion can lead to formation of materials with interesting nontoxic and differentiated properties [125].

Sharma et al. [129] produced composites of starch with PVA with the introduction of cellulose fibers and PEG. The cellulose fibers were obtained from the rice straw; this approach allows one to utilize agricultural wastes and produce biodegradable materials. Comprehensive study of the composition properties showed that the application of cellulose fibers and PEG improved the compatibility between starch and PVA, the mechanical characteristics up to 52%, enhance the water vapor permeability. Debnath et al. [130] used microcrystalline cellulose synthesized from elephant grass (growing in the North-Eastern states of India) to reinforce the starch-based materials obtained by solution-casting technique. The overall enhancement in rigidity, mechanical, and thermal properties was observed due to the good compatibility between molecules of microcrystalline cellulose and starch matrix. Mixing cassava starch with bacterial cellulose and pullulan led to increased elasticity, along with thermal stability and decreased water vapor permeability. Therefore, this material consisting of only natural polysaccharides is a potential packaging item [131].

One more natural polymer class actively used to produce the biodegradable packaging materials are proteins. Different proteins were mixed with starch for this aim [132,133,134,135,136,137]. Fonseca Ferreira et al. [132] obtained blends of cassava starch with soy protein at different concentrations via extrusion method. Varying the protein content, one increased the elastic modulus by 120% and decreased the water vapor permeability by 25% (at 60 wt% of protein) and oil permeability compared to neat starch. Thus, such blends can be considered for packaging for foods with high lipid contents.

Of special interest are the materials based on starch with essential oils [138,139,140]. Ghasemlou et al. [138] introduced two essential oils (Zataria multiflora Boiss or Mentha pulegium) to improve mechanical properties, water vapor permeability, as well as to impart antimicrobial activity (effective inhibiting the growth of *E. coli* and *S. aureus*) and preserve the material transparent with slight yellowness. Varying the concentrations of these essential oils (1–3% (*v*/*v*)), one could select the compositions with optimal values of tensile strength and elongation at break, oxygen barrier properties, etc. Thus, the presence of oils in the composition with starch allows one to propose the formulation of antimicrobial biodegradable films for various food packaging applications.

Another new trend in the development of biodegradable and active antioxidant materials is based on the introduction of plant extracts, for example, thyme extract [141], rosemary extract [142], and so on. The plant extracts containing natural polyphenols were successfully incorporated into the polysaccharide matrix. The presence of the plant extract improved the UV-blocking properties of the starch-based films and accelerated biodegradation (tested in compost) [142].

### 2.3. Chitin and Chitosan

Chitin is the second abundant polysaccharide in nature after cellulose. The most widespread derivative chitosan is produced by its deacetylation. The sources and production of chitin and chitosan, their chemical structure, and their properties have been studied extensively, for example [143,144,145]. Thanks to their availability, biocompatibility, biodegradability, and many other advantageous properties, chitin and chitosan have been used in many application fields (medicine, water purification, agriculture, cosmetic, food, etc.). Since the early works until the mid-1990s, these polysaccharides have been predominantly applied for biomedical purposes, in particular as drug carriers [146,147], in tissue engineering [148,149,150], and as absorbents, especially for heavy metals [151,152,153], etc.

The mixing of chitin and chitosan with synthetic polymers was one the first approaches to the development of biodegradable packaging materials based on them. Biodegradable compositions based on these polysaccharides first mentioned in the 1990s were produced with polyethylene [27,154,155,156]. At 10 wt% content of polysaccharide, the maximal rate of biodegradation in soil was demonstrated by chitin-based systems [154]. Later in [155], compositions of chitin and chitosan with low-density polyethylene were produced in a rotor disperser. However, in this case, the maximal content of polysaccharide was increased up to 50 wt% and better biodegradability was found for chitosan-based systems. One more polyolefin, namely polypropylene, was mixed with chitin nanowhiskers (maximal concentration 10 wt%) [157]. Their thermal, barrier, mechanical, and rheological characteristics have been thoroughly analyzed to propose them as alternatives to existing industrial products with synthetic additives for packaging.

Combining chitosan with poly(vinyl alcohol) representing a synthetic non-toxic and water-soluble polymer is a promising approach for the production of biodegradable packaging materials [158,159,160,161,162,163,164]. It has been demonstrated that hydrogen bonding between –OH group of poly(vinyl alcohol) and –NH_2_ group of chitosan takes place. It is declared that these polymers exhibit good compatibility and homogeneous films are formed [159]. PVA introduction increases tensile strength and elongation at break of the biodegradable materials, while preserving antioxidant and antibacterial capacities.

The development of biocomposites based on chitin/chitosan with naturally occurring polymers is another fruitful research field. These works include those with polyesters which can serve as an alternative to non-biodegradable synthetic polymers [52,53,54,165,166,167,168,169,170,171,172,173,174,175,176,177,178,179]. Authors in [171,173] showed that crystallization of polycaprolactone is suppressed by blending (solution-cast technique) with chitin/chitosan, and the hydrogen bonding between these polymers was established (using DSC, FTIR, and ^13^C NMR). The series of works [52,53,54] showed the possibility to produce plasticized PLA-chitosan compositions via solid-phase mixing under the action of shear deformations. The results showed a decrease in the elastic modulus and tensile strength and a rise in the elongation at break, along with biodegradability.

Another branch of investigations is the production of polysaccharide–protein compositions since both components are natural polymers. Therefore, in [180] it was proposed to mix chitosan with sodium caseinate plasticized with glycerol as perspective edible films; their water vapor permeability was explored. Many scientific groups consider gelatin to be a promising polymer for mixing with chitin/chitosan [181,182,183,184,185,186,187]. Sahree et al. [185,186] thoroughly explored the characteristics of gelatin-chitin nanocomposite (with and without corn oil) in terms of physicochemical (mechanical, thermal, optical) and antifungal properties. The optimal concentration of the chitin nanoparticles in gelatin matrix as well as optimal content of corn oil to create biodegradable materials with more acceptable properties for food packaging was outlined. Hosseini et al. [182,183] used fish gelatin from fish skin—a by-product of the fish-processing industry. The introduction of chitosan and chitosan nanoparticles caused an increase in the tensile strength and elastic modulus, but a fall in water vapor permeability compared to neat gelatin.

Hai et al. [188] developed biocomposites prepared by mixing chitin nanofibers and bamboo cellulose nanofibers. Full biodegradation within a week and good mechanical (rise in tensile strength and elastic modulus up to 3 and 1.3 times, respectively, as the concentration of cellulose nanofibers increases) and thermal properties (better thermal stability than that for pure cellulose nanofibers) make this material promising for possible applications as food packaging. Wu et al. [189] presented a facile method to graft chitosan onto the oxidized cellulose films. The resulting material will possess high transmittance, barrier properties, and antimicrobial properties, combining advantages of each polysaccharide.

The introduction of oils (bioactive compound) into polysaccharides is the first step in the production of active packaging materials to improve safety, maintaining quality and extending the shelf life of products. In [190], local buriti oil (Mauritia flexuosa L.f.) was introduced into chitosan films, which resulted in the production of flexible films with antimicrobial properties and improved water vapor barrier. A recent review [191] discusses all features and advantages of production and application of chitosan–essential oil compositions. These systems are mainly used in food preservation coatings, food packaging materials, and antimicrobial agents of foodborne pathogenic bacteria.

According to the literature reviewed, interest in chitin and chitosan has arisen in the past 10–15 years. This indicates that the potential for application of these polymers still needs to be explored.

### 2.4. Some Other Polysaccharides

This section could be extremely large, but this review presents only some polysaccharides with potential application in packaging industry. For example, pullulan polysaccharide obtained from microorganisms has been attracting more attention in view of different application fields, including packaging, in recent years. Rai et al. [192] discussed the possibility of using pullulan thanks to its water-soluble and non-toxic nature. Shanmugan et al. [193] reported that pullulan/graphene biocomposite coated on nanocellulose film improved the surface, barrier, and antimicrobial properties of the latter. In new research [194], the authors described the features of production of pullulan/paper/zein laminates as promising food packaging material. Another polysaccharide, hyaluronic acid, is known as a polymer used mainly in medicine, cosmetics, pharmaceutics, etc. However, some attention to this polymer in the light of application as a packaging material was demonstrated in [195]. Oliveira et al. [196] paid attention to arabinogalactan (in work ora-pro-nobis mucilage) as a promising alternative to traditional non-biodegradable plastics.

In the author’s opinion, the polysaccharides listed below are primarily of interest in terms of food preservation and prolonging food storage, but to a lesser degree to substitute the petroleum-based packaging materials.

#### 2.4.1. Pectin

Pectin is a heteropolysaccharide mainly presented in fruits and vegetables. It is used as a food additive thanks to its stabilizing and gelling ability. Pectin dissolves in water and is insoluble in organic solvents. The most often used plasticizer for pectin is glycerol [197]. Since this plasticizer is often used with starch, that is why these polysaccharides are often mixed (see Section 2.2). Citrus pectin mixed with proteins (fish skin gelatin) showed good potential as packaging material [198,199]. Liu et al. [198] demonstrated that the resulting materials are flexible, with decreased water solubility and water vapor transmission rate compared to those of neat pectin.

Pereira et al. [200] first reported the pectin-based biodegradable material with *Salicornia ramosissima* (representing a halophyte plant very rich in sodium, magnesium, potassium, calcium, and manganese with antioxidant properties). Comprehensive investigation of their properties showed that these flexible and opaque materials degrading in seawater and soil can be potential eco-friendly packaging materials.

Essential oils as active additives are often incorporated into the pectin matrix to produce functionalized materials: clove essential oil [201], marjoram essential oil [202], rosemary oil [203], neem oil [204], etc. Akhter et al. [203] have demonstrated the synergic effect between mint and rosemary essential oils and nisin using different polysaccharides, namely chitosan, starch, and pectin blends. It was concluded that the introduction of rosemary oil and nisin improved the water barrier properties, tensile strength, and thermal stability of the biocomposites. Besides, the high antimicrobial effect against some pathogenic strains was found. Other components often introduced into pectin are extracts derived from plants, for example, sage leaf extract [205]. The introduction of sage leaf extract led to an increase in antioxidant ability, while thermal stability and water resistance decreased. Overall, 0.6 wt% of extract resulted in a rise in tensile strength and elongation at break. The incorporation of extract led to a rise in yellowness; as its content increased, the film became opaque as a result of better light barrier property. Such compositions were proposed to be potentially used for food-based applications.

#### 2.4.2. Alginate and Carrageenan

Both alginate and carrageenan polysaccharides are obtained from algae. They are widely used in the food industry as food additives, mainly as thickeners and stabilizers [206], medicine [207,208,209], etc. However, over the years, some investigations have been dedicated to the development of composites based on these polysaccharides for packaging industry [210,211]. Therefore, in the investigation of the present year [200], a group of scientists from China produced complex material based on alginate, carragenan, and shellac with cellulose nanocrystals for reinforcing, obtained from enteromorpha. This approach allows one to solve the problem of accumulation of *Enteromorpha prolifera* (main pollutant of “green tide”), which was used for the production of cellulose. The food storage experiment has established that this material is excellent for preserving fresh food. Carneiro-da-Cunha et al. [212] have coated PET film by nanolayers of sodium alginate and chitosan (five layers) and investigated their water–vapor permeability and thermal and mechanical properties. The results allow one to conclude that these materials can be promising multilayer edible coatings with enhanced mass transfer and mechanical properties. One more polysaccharide mixed with alginate is pectin [213,214]. Makaremi et al. [214] developed alginate–pectin films incorporated with ascorbic and lactic acids. Thanks to low cost, good mechanical characteristics, and antibacterial properties, this composition showed potential to be used in food packaging. As a trend of recent years, alginate was also mixed with oregano essential oil [215]. Different degrees of cross-linking by calcium carbonate and effect of oil demonstrated changes in characteristics such as mechanical, optical, antibacterial, and water vapor barrier ones. The minimum concentration of oil provided antibacterial efficacy. Theagarajan et al. [216] reviewed the main approaches in the development of alginate-based food packaging materials.

Interest in carrageenan in terms of the development of new biodegradable packaging materials has arisen in the last decade. Carrageenan is commonly used as edible films for preservation of food [217,218] and packaging materials for keeping fresh food (hold texture and aroma, prevent discoloration) [219,220]. Most often, carrageenan is mixed with cellulose/its derivatives [221,222,223,224], starch [225,226,227,228], essential oils [223,229], etc. Castaño et al. [230] produced carrageenan/starch and carrageenan/(faba bean) protein compositions, then their thermal, mechanical, and antifungal properties, as well as structures, were thoroughly explored. The results revealed that their mechanical properties are in the range known for synthetic polymers used for food packaging. The essential oils introduced can be considered as antifungal agents. The materials can be considered as the biodegradable alternatives to packaging materials in the food industry. Nur Fatin Nazurah R. and Z. A. Nur Hanani [229] demonstrated that the introduction of plant oils into carrageenan resulted in a reduction in the moisture content and solubility in water and enhancement in the water vapor permeability. The best characteristics were observed for carrageenan–olive oil compositions.

## 3. Conclusions

The research of the biodegradable plastics based on natural polymers (polysaccharides) must be developed toward the direction of reducing production costs and improving performances (mechanical, water vapor permeability, O_2_/CO_2_ barrier, thermal, optical, antibacterial properties, and biodegradability). According to the literature analysis, the potential of polysaccharides is still very high, since they are obtained from renewable sources, non-toxic, and fully biodegradable. The development of specific polysaccharide-based materials is also defined by the geographical aspect. It is obvious that the interest in creation of blends based on polysaccharides with synthetic/other natural polymers will keep on rising, since this approach allows for developing utilization techniques for non-degradable components. Besides, the development of any future trends depends on the possibility to produce material on a large scale. This aspect decelerates the progress and manufacturing application. However, the science community has proposed new approaches in this field. The application of essential oils attracts a lot of attention, and it is promising. The application of essential oils in the systems based on polysaccharides allows one to impart antibacterial and antioxidant properties. One more interesting direction is the production of “smart materials”. Such systems can be light-, ammonia-, and pH-sensitive. Despite the approaches described in the work, there are many directions for the development of biodegradable packaging materials based on polysaccharides, including their cross-linking, introduction of metals, metal oxides, clays, etc. However, there is a large potential for development of new formulations to create a safe, sustainable, and pollution-free environment for current and future generations.

## Data Availability

Not applicable.

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
