# Peer review of "Polysaccharides for Biodegradable Packaging Materials: Past, Present, and Future (Brief Review)"

_polymers, 2023, doi:10.3390/polym15020451_

Round 1
Reviewer 1 Report
This manuscript is organized towards polysaccharides for biodegradable packaging and presents a comprehensive report of composites of cellulose, starch, chitin and their derivatives with several other examples of polysaccharides with man-made polymers, natural polymers and small molecules. In general, the authors expresses their concern for the key properties of the composites such as mechanical properties, thermal properties, water vapor permeability, and antimicrobial effect in each group of investigations, and also cites exhaustive and adequate references. However, there are still some issues that need to be addressed.
1. In the introduction section, soybean polysaccharides and degradable pigments, which are especially presented, actually occupy little space in the text, while at the same time, several polysaccharides, which are mainly introduced in the text, are not succinctly highlighted in the introduction.
2. This reviewer believes that in order, 2.3. some other polysaccharides should be labeled 2.4, as there already exists section 2.3. chitin and chitosan also, subheadings 2.3.1. pectin and 2.3.6. alginate and carrageenan under this section are ordered incorrectly.
3. In the presentation of the individual polysaccharide composites, there is a lack of attention to the optical properties of the material, which is one of the core properties of the packaging material.
4. One table to summarize and compare the advantages and disadvantages of different polysaccharides should be provided for better understanding of readers.
5. As a review article, some recent and highly relevant articles should be included, such as:
1) Packaging: Nanomaterials 12 (18), 3158, 2022; Nanomaterials 10 (1), 150, 2020; Biobased materials for food packaging; etc.
2) Chitosan: Recent advancements in applications of chitosan-based biomaterials for skin tissue engineering; New Ulva lactuca Algae Based Chitosan Bio-composites for Bioremediation of Cd(II) Ions; Sources, production and commercial applications of fungal chitosan: A review; etc.
3) PVA/Gelatin: Packaging and degradability properties of polyvinyl alcohol/gelatin nanocomposite films filled water hyacinth cellulose nanocrystals
4) Etc.
6. Please carefully check the whole manuscript. There are still some typos and grammar issues. In addition, please carefully check the references to ensure the full information is included.
7. One more perspective section is suggested to show the challenges and future solutions to guide the future studies in this field.
Author Response
Dear Reviewer, thank You for careful reading of the manuscript and made remarks.
Question 1. In the introduction section, soybean polysaccharides and degradable pigments, which are especially presented, actually occupy little space in the text, while at the same time, several polysaccharides, which are mainly introduced in the text, are not succinctly highlighted in the introduction.
Answer 1. The introduction section "introduces" the readers to the idea of creation of the biodegradable materials based on polysaccharides. Some common and some interesting aspects are discussed (the corresponding references are given for readers). But the main body of the text contain information about the most popular and some other polysaccharides and their systems with components listed in the end of the introduction section. This is one of the reasons why the work is in format of brief review.
Question 2. This reviewer believes that in order, 2.3. some other polysaccharides should be labeled 2.4, as there already exists section 2.3. chitin and chitosan also, subheadings 2.3.1. pectin and 2.3.6. alginate and carrageenan under this section are ordered incorrectly.
Answer 2. The section numbers have been changed. Thank You very much for Your careful reading.
Question 3. In the presentation of the individual polysaccharide composites, there is a lack of attention to the optical properties of the material, which is one of the core properties of the packaging material
Answer 3. Some discussions in terms of optical properties have been introduced into the text now.
Question 4. One table to summarize and compare the advantages and disadvantages of different polysaccharides should be provided for better understanding of readers.
Answer 4. The table is given in the text now.
Question 5. As a review article, some recent and highly relevant articles should be included, such as:
1) Packaging: Nanomaterials 12 (18), 3158, 2022; Nanomaterials 10 (1), 150, 2020; Biobased materials for food packaging; etc.
2) Chitosan: Recent advancements in applications of chitosan-based biomaterials for skin tissue engineering; New Ulva lactuca Algae Based Chitosan Bio-composites for Bioremediation of Cd(II) Ions; Sources, production and commercial applications of fungal chitosan: A review; etc.
3) PVA/Gelatin: Packaging and degradability properties of polyvinyl alcohol/gelatin nanocomposite films filled water hyacinth cellulose nanocrystals
4) Etc.
Answer 5. According to Your remark some references have been cited.
Question 6. Please carefully check the whole manuscript. There are still some typos and grammar issues. In addition, please carefully check the references to ensure the full information is included.
Answer 6. The whole manuscript was carefully re-read and checked in order to improve it.
Question 7. One more perspective section is suggested to show the challenges and future solutions to guide the future studies in this field.
Answer 7. Corresponding information is presented in the Conclusions section now.
Reviewer 2 Report
This manuscript in this form, even as a mini review is not going to add any knowledge to valuable information for the researchers of this field and it needs substantial improvement with thorough rewriting. Some of the main concerns include:
- The abstract and conclusion are primitive and will not encourage the reader to go through the manuscript.
- The advantages and drawbacks of the addressed polysaccharides had to be fully addressed.
- Some figures and tables should be provided.
- Line 70, a paragraph on cellulose and its derivatives as packaging materials is super insufficient even for a mini-review. This manuscript could be strengthened to some extent by the following papers:
https://doi.org/10.1016/j.carbpol.2021.118550; https://doi.org/10.1016/j.carbpol.2022.119910; https://doi.org/10.1016/j.foodcont.2022.109595
- Line 120, needs justifications.
- A subheading named “suggestions to improve interface/interaction of natural/synthetic polymers” with profound discussions is necessary.
- Future trends and prospects must be provided.
Author Response
Dear Reviewer, thank You for careful reading of our manuscript and made remarks.
Question 1. The abstract and conclusion are primitive and will not encourage the reader to go through the manuscript.
Answer 1. The abstract and conclusion have been rewritten.
Question 2. The advantages and drawbacks of the addressed polysaccharides had to be fully addressed"
Answer 2. Corresponding table with this information is presented in the text now.
Question 3. Some figures and tables should be provided
Answer 3. The table and scheme are presented in the introduction section now.
Question 4. Line 70, a paragraph on cellulose and its derivatives as packaging materials is super insufficient even for a mini-review. This manuscript could be strengthened to some extent by the following papers:
Answer 4. These interesting works listed by the Reviewer were added into the Introduction section.
Question 5. Line 120, needs justifications.
Answer 5. The corresponding explanations are presented in the text now.
The mechanical properties and transparency were taken as the comparative parameters: composites with TEMPO oxidized cellulose nanofibers showed the better results. Namely, their transparency comprised 56%, while for neat carboxymethyl cellulose it was 65%. Such high value was explained by the smaller size and higher homogeneity of TEMPO-oxidized nanofibers. Higher mechanical characteristics were explained by less dimension and higher specific surface area for TEMPO-oxidized compared to other types of cellulose nanomaterials [59].
Question 6. A subheading named “suggestions to improve interface/interaction of natural/synthetic polymers” with profound discussions is necessary
Answer 6. Undoubtedly, this topic is very important and necessary in the description of systems based on natural and synthetic polymers. This aspect can serve as a basis for independent huge review work. However, within the framework of this article, it was not possible to cover this topic, since the emphasis in the work is aimed at reviewing compositions not only with synthetic polymers. The author is grateful to the referee for such a high interest in the work and for such valuable advice.
Question 7. Future trends and prospects must be provided
Answer 7. Relevant text about future trends and prospects is provided in the conclusions section now.
Reviewer 3 Report
The whole manuscript is well and clearly written – a well-designed review of the recent findings. I have some comments/questions that can improve this short review:
1. The problem with the characteristic of polysaccharides and repeatability of the e.g. molecular mass and degree of deacetylation (chitosan) should be mentioned in the future application of these materials on a large scale.
2. What are the commonly applied ratios between synthetic and natural components?
3. In general: do described mixtures exhibit similar properties (important for the packaging industry sector) to the currently used polymeric packaging? The general drawbacks of polysaccharides for this purpose should be mentioned.
4. Please give a short note regarding the current packages being produced and containing polysaccharides, if there are any.
5. Others: Line 282 (lack of dot), Line 337 (deceased), Line 354 (0,6 wt% - should be 0.6 wt%)
Author Response
Dear Reviewer, thank You for careful reading of our manuscript, made remarks. I am grateful for the appreciation of the work.
Question 1. The problem with the characteristic of polysaccharides and repeatability of the e.g. molecular mass and degree of deacetylation (chitosan) should be mentioned in the future application of these materials on a large scale.
Answer 1. Thanks for such an advice, some corresponding information is presented in the conclusions section now.
Question 2. What are the commonly applied ratios between synthetic and natural components?
Answer 2. According to the literature data the most preferrable content of the natural component in the systems with synthetic polymers is about 10-20 wt%. It is caused by several reasons, including preservation of good mechanical properties. However, there are some investigations describing the compositions containing up to 60 wt% of natural polymers. For example, my colleagues and I developed compositions based on LDPE with different polysaccharides, as well as with PLA and starch [for example, 27–35; 51–54; 95–99 from the manuscript list of references].
Question 3. In general: do described mixtures exhibit similar properties (important for the packaging industry sector) to the currently used polymeric packaging? The general drawbacks of polysaccharides for this purpose should be mentioned.
Answer 3. This information is given in the table now.
Question 4 Please give a short note regarding the current packages being produced and containing polysaccharides, if there are any.
Answer 4. Some information about commercial packaging material is presented in the Table now.
Question 5. Others: Line 282 (lack of dot), Line 337 (deceased), Line 354 (0,6 wt% - should be 0.6 wt%)
Answer 5. The required changes have been made.
Round 2
Reviewer 1 Report
Authors have addressed all the issues well. An acceptance is suggested.
Reviewer 2 Report
Dear Editor,
Based on the changes made to the manuscript, I believe that this work is suitable for publication.
Best regards,
Hossein